# Liquid Biopsy in Pediatric Renal Cancer: Stage I and Stage IV Cases Compared

**DOI:** 10.3390/diagnostics10100810

**Published:** 2020-10-12

**Authors:** Elisabetta Rossi, Angelica Zin, Antonella Facchinetti, Cristina Poggiana, Lucia Tombolan, Maria Carmen Affinita, Paolo Bonvini, Luisa Santoro, Francesca Schiavi, Gianni Bisogno, Rita Zamarchi

**Affiliations:** 1Department of Surgery, Oncology and Gastroenterology, Oncology Section, University of Padova, Padua, Italy; elisabetta.rossi@unipd.it (E.R.); antonella.facchinetti@unipd.it (A.F.); 2Veneto Institute of Oncology IOV—IRCCS, Padua, Italy; cristina.poggiana@iov.veneto.it (C.P.); francesca.schiavi@iov.veneto.it (F.S.); 3Institute of Pediatric Research, Fondazione Città della Speranza, Padua, Italy; angelica.zin@unipd.it (A.Z.); lucia.tombolan@unipd.it (L.T.); paolo.bonvini@unipd.it (P.B.); 4Department of Woman’s and Children’s Health, Hematology and Oncology Unit, University of Padua, Padua, Italy; draffinita@gmail.com (M.C.A.); gianni.bisogno@unipd.it (G.B.); 5University Hospital of Padova, Institute of Pathology, Padua, Italy; luisa.santoro@aopd.veneto.it

**Keywords:** pediatric renal cell carcinoma (RCC), case report, circulating tumor cells (CTCs), circulating endothelial cells (CECs), liquid biopsy

## Abstract

Pediatric renal cancer is rare, and robust evidence for treatment recommendations is lacking. In the perspective of personalized medicine, clinicians need new biomarkers to improve risk stratification and patients’ follow-up. Herein, we analyzed some liquid biopsy tools, which have been never tested in pediatric renal cancer: namely, circulating tumor cells (CTCs); the expression of M30, an apoptosis marker, to test CTC metastatic potential; and c-MET expression in CTCs, because of its role in renal cancer progression and drug-resistance. Furthermore, we evaluated the Circulating Endothelial Cells (CECs), whose utility we previously demonstrated in adult metastatic renal cancer treated with anti-angiogenic therapy. We compared two renal cell carcinomas of clear-cell type, stage I and IV, which underwent surgery and surgery plus Sunitinib, respectively. Baseline CTC level and its changes during follow-up were consistent with patients’ outcome. In case 2, stage IV, the analysis of CECs performed during Sunitinib revealed a late response to treatment consistent with poor outcome, as the finding of M30-negative, viable cells. Noteworthily, few CTCs were MET-positive in both cases. Our study highlights the feasibility for a change in the prognostic approach and follow-up of childhood renal cancer, with a view to guide a better treatment design.

## 1. Introduction

Pediatric renal cancer is a rare tumor, affecting 8.8 children per million per year in Europe [1]. Wilms tumor accounts for 90% of kidney cancers, whilst renal cell carcinoma (RCC) covers only 1% of the total cases [2]. To date, RCC patients have a 90% survival rate for stage I and II, which progressively declines to 30% for stage IV disease [3]. 

RCC rarity represents the main challenge to obtain robust evidence for treatment recommendations. So far, treatment options include surgery, eventually combined with adjuvant treatment [4], whilst retroperitoneal lymphadenectomy is still a matter of debate [5]. In case of unresectable or metastatic disease current adult guidelines, which include cytokines and targeted drugs, are adopted, and, despite RCC tumor biology differing between childhood and adulthood disease, some encouraging successes have been reported, mainly with Sunitinib [6].

In consideration of the dismal prognosis of the disseminated disease, the possibility to identify patients with apparent localized disease that may benefit from personalized treatments, underline the need for new biomarkers to improve risk stratification and follow-up of these patients [6]. 

Nowadays, liquid biopsy has emerged as a minimally invasive source of tumor cells over the course of the disease, even when tumor or metastasis material is not available or when classical biopsy has unacceptable risks for the patient. Liquid biopsy for cancer patients now includes novel biomarker classes: namely, circulating tumor cells (CTCs), circulating-free tumor DNA (ctDNA), tumor derived extracellular vesicles (tdEV), and exosomes, which all permit a more comprehensive view of the circulating compartment. 

Among these, the role of CTCs in metastatic disease was first accepted [7], since the technological advances over the last fifteen years have consented to the quantification of CTCs in the peripheral blood of epithelial cancer patients. Furthermore, inverse correlations between CTCs and progression-free (PFS) and overall survival (OS) have been reported in metastatic cancer of the breast [8], colon-rectum [9], and prostate [10]. Therefore, the Food and Drug Administration approved the In Vitro Diagnostic (IVD) use of the CellSearch (CS) CTC assay in these three malignancies. Moreover, exploiting the same technology, CTCs have been reported in several different solid tumors [11] and are related to patient outcomes.

On these bases, we chose to apply in pediatric RCC, the enumeration of CTCs, which we had successfully exploited in adult renal cancer patients as a prognostic marker [12]. Furthermore, we assessed in parallel Circulating Endothelial Cells (CECs), whose utility we previously demonstrated in adult metastatic RCC, treated with anti-angiogenic drug [13]. 

Herein, we compare two RCC cases—stage I and IV, respectively—who underwent surgical and medical intervention depending on their disease stage, providing for the first time the proof of principle that CTCs and CECs are actionable tools in pediatric RCC. 

## 2. Patients and Methods

### 2.1. Patients’ Clinical Characteristics

#### 2.1.1. Informed Consent

Parents of subjects recruited in this study signed the informed consent forms and consented to publish anonymized patient data. The study outcome did not affect patient management and will not affect the future management of the patient who was censored at the end of this study. 

The Ethics Committee of the institution within which the work was undertaken has approved the protocol for this research project. It conforms to the provisions of the Helsinki Declaration as revised in 2013 (CE IOV: 2012/52, 24 September 2012).

#### 2.1.2. Case 1—Presentation

A 15-year-old girl with an unremarkable medical history underwent pelvic ultrasound due to amenorrhea with evidence of a 4.5 × 5.4 cm hypoechoic lesion in the superior part of the left kidney. The patient was transferred to our hospital and an MRI—performed in September 2017—confirmed the renal lesion. No lung lesions were evident on CT scan.

In October 2017, a needle biopsy confirmed the clinical suspicion of renal carcinoma; therefore, a tumor resection with left nephrectomy was performed. 

Histology on the surgical specimen confirmed the diagnosis of renal cell carcinoma of clear-cell type (ccRCC), with surgical margins free of neoplastic invasion. Genomic analysis for germ-line mutations of *FH*, *MET*, *PETEN*, *SDHB*, and *VHL* resulted negative.

No further treatment was administered, and the child is still alive and in complete remission. 

#### 2.1.3. Case 2—Presentation

A 12-year-old girl presented at the emergency department of our Hospital after two episodes of macro-hematuria. A complex cystic lesion in the lower part of the left kidney was revealed by abdominal ultrasound. A few days later, a CT scan demonstrated a solid 4.4 cm × 4.2 cm × 5.5 cm lesion in the lower half of the left kidney, well vascularized and inhomogeneous due to the presence of thin lamellar calcifications. Lymph node conglomerates were evident in the homolateral periaortic area for an extension of 10 cm and axial diameters of 3.7 × 2.8 cm. Multiple bilateral lung lesions were also evident. 

In May 2017, a tumor resection with nephrectomy and lymphadenectomy was performed. The diagnosis was renal cell carcinoma of clear-cell type (ccRCC), with nodal invasion. Genomic analysis for germ-line mutations of *FH*, *MET*, *PETEN*, *SDHB* and *VHL* resulted negative.

After post-surgical recovery, at the end June 2017, the patient underwent treatment with Sunitinib, 15 mg/m^2^/dose, orally administered according to a 4-weeks-on + 2-weeks-off treatment schedule [14]. 

Three months later (early October), at the end of the third Sunitinib cycle, a reduction of the lung lesions was documented on CT scan, and Sunitinib treatment was continued. However, 2 months later (early December 2017), the child was urgently admitted with severe headache and vomiting, and a 4.5 cm cerebral metastasis found on MRI was resected. A new CT scan performed 1 month later (early January 2018) showed disease progression, both in the lungs and in the abdomen.

The child eventually died 10 months from diagnosis.

### 2.2. Total, M30-Positive and MET-Positive CTC and CEC Enumeration

We quantified CTCs and CECs in whole blood with CellSearch System (CS) (Menarini, BO, Italy) [8,15]. According to manufacturer’s instructions, an event is classified as CTC or CEC when its morphological features are consistent with that of a cell and it exhibits the phenotype EpCAM+, CK+, DAPI+ and CD45− or CD146+, CD105+, DAPI+, and CD45−, respectively. Quantitative results were expressed as per 7.5 mL peripheral blood (PB) and per 4 mL PB for CTC and CEC, respectively.

To investigate drug-induced cell death over Sunitinib treatment, we detected viable and apoptotic CTCs, using the CS assay in conjunction with anti-M30 monoclonal antibody (mAb). M30 is a neoepitope disclosed by caspase cleavage at cytokeratin 18 in early phases of apoptosis [16]; results are expressed as the total number of CTC and M30-positive CTC per 7.5 mL PB.

Likewise, in some samples, we used the CS assay with the FITC-conjugated c-MET mAb (Clone SP44, SpringBioscience, Pleasanton, CA 94588, USA) to recognize the *c-MET* gene product, a transmembrane receptor-like protein with tyrosine kinase activity; results are expressed as the total number of CTC and MET-positive CTC per 7.5 mL PB.

### 2.3. Genomic Analysis

Genomic DNA was extracted from 10 mL samples of PB leukocytes. We performed sequencing analysis for mutations in *FH, MET, PETEN, SDHB* and *VHL* genes.

## 3. Discussion and Conclusions

In both case 1 and 2, histology revealed ccRCC, the most common among rare cancers in adolescents; disease stage was I and IV, respectively. 

After diagnosis, both children underwent radical nephrectomy, the first treatment option for RCC [17] in childhood; retroperitoneal lymphadenectomy was included in case 2, because of documented lymph node invasion. After surgery, this last patient, suffering stage IV ccRCC, then underwent repeated cycles of Sunitinib, according to treatment schedule 4-weeks-on, 2-weeks-off. This treatment showed only limited disease control as disease free interval and transient reduction of lung lesions. 

Surgical intervention with curative intent was possible only in case 1, stage I disease, and complete remission was achieved, whilst after the onset of brain metastasis, we documented an ominous outcome for the case 2, stage IV disease.

Table 1 compare clinical-pathological characteristics, treatments and imaging reports throughout the continuum of the care, with findings obtained through liquid biopsy.

The main finding of our study is that we were able, for the first time to our knowledge, to detect CTCs in childhood RCC. Notably, in case 2, stage IV disease, which showed worst outcome, we found at baseline 3 CTCs/7.5 mL PB, a level that we previously reported prognostic of significantly shorter PFS and OS in adult RCC [12]. Conversely, we found only 1 CTC in the post-operative sample of case 1, who is still in complete remission.

Moreover, as previously reported in adult RCC [13], CTC levels changed over the course of treatment. Notably, in case 1, the CTC number turned negative, whilst in case 2, it changed from positive to negative and then back to positive over the follow-up period. 

These changes mirrored the clinical conditions of the two patients. In fact, in case 1, CTCs turned undetectable when the patient was in complete remission. Conversely, in case 2, we found some CTCs in all but one sample, suggesting that only partial disease control was reached. 

Similarly, the lack of the apoptosis marker M30 at the last CS test of case 2 indicates an aggressive disease, consistent with documented lung and abdomen progression. Indeed, when we applied here the parameter ΔAUC [13] that expresses the balance between viable (M30-negative) and apoptotic (M30-positive) cells, over the Sunitinib treatment, we obtained a positive result (ΔAUC = 499). Noteworthily, in adult ccRCC treated with Sunitinib, a positive value of ΔAUC was associated with higher probability of disease recurrence for distance metastasis [13].

Furthermore, the evaluation of synchronous detection of CTCs and CECs in case 2 over the course of anti-angiogenic treatment also confirmed in childhood disease what we previously observed in mRCC treated adult subjects [13], in whom a delayed response of CECs to Sunitinib was related to higher CTC values and treatment failure. 

Indeed, here we found only a limited amount of CECs at the end of the 1st cycle of Sunitinib (91 cells), which dropped down to normal levels at the end of the 4th cycle of anti-angiogenic treatment (10 cells); at both time points, CTCs were also detectable (1 and 2 CTCs, respectively). Conversely, the only CTC-negative test was synchronous with the finding of 323 CECs/4 mL PB, reached only at a later time-point of follow-up. Since the response to Sunitinib is delayed, it is conceivable that tumor evasion might occur because of the on-off treatment schedule, as we previously reported in adult ccRCC [13]. In any case, the regression analysis of synchronous detections showed a strong exponential inverse correlation between CTCs and CECs (*cc* = −0.987), further confirming that the anti-endothelial targeting and the anti-tumor cell effects are strictly linked in vivo.

Finally, the expression of MET in at least a few CTCs in both cases deserves consideration. Indeed, in both cases the genomic analysis for germ-line mutations of genes noted for involvement in RCC did not show *MET* gene alterations. This finding is not surprising, since c-MET over-expression is rare in primary cancer, whilst it is known to confer metastatic potential to cancer cells in several malignancies, including renal cell carcinoma (RCC). Moreover, Zhang T. and colleagues have already demonstrated that CTCs with c-MET amplification could be detected in patients with gastric, colorectal, and renal cancers [18]. Noteworthily, MET has been implied in a common mechanism of resistance to targeted drugs, which includes EGFR and VEGFR inhibitors [19], and can be successful contrasted by Cabozantinib in the second-line setting [20]. Hence, it is intuitive that detecting c-MET-positive CTCs in candidates for Sunitinib treatment is clinically relevant to design better treatment strategies.

Our study shows some limits, implicit of case-reporting studies, namely: few patients and treatments, and follow-up and timeline schedule of blood draws were nonhomogeneous because of different disease stages. However, we think it offers the rationale for further investigations of liquid biopsy tools in larger pediatric RCC cohorts, mainly in the view of employing personalized treatment.

In conclusion, we have shown for the first time that the enumeration of CTCs is actionable in pediatric RCC. The changes of this parameter mirror the disease condition, while the expression of an apoptosis marker or a product of genomic alteration on CTCs gives further details about the metastatic potential of the disease and ongoing drug-resistance. Then, over the course of anti-angiogenic treatment, the synchronous detection of CTCs and CECs allow for disclosing its effectiveness in individual patients. 

Pediatric RCC is a rare tumor and the clinical utility of liquid biopsy in this malignancy warrants further confirmation in larger studies. However, we think that the cases discussed here are paradigmatic and highlight the need for a change in prognostic approach and follow-up of childhood RCC, in view to guide a better treatment design.

## Figures and Tables

**Table 1 diagnostics-10-00810-t001:** Clinical diary of treatments, imaging and pathological evaluations, and liquid biopsy data.

*pt*	*Date of Diagnosis*	*Date of Follow-Up/Treatments*	*Blood Draw Date*	*CTCs/7.5 mL*	*M30+/Sample*	*MET+/Sample*	*CECs/4 mL*	*Treatments*	*IMAGING/Pathological Evaluation*	*Report*	*Objective Response*
***1***	19-Sep-17								***MRI***	***renal lesion***	
	5-Oct-17								***needle biopsy***	***ccRCC***	
		13-Oct-17						***nephrectomy***			
			3-Nov-17	**1**	**nd**	**1**	**28**				
			21-Feb-18	**0**	**nd**	**0**	***nd***				***CR***
***2***			5-May-17	***3***	***3***				***CT scan***	***renal lesion 4.4 cm × 4.2cm × 5.5 cm; LN and lung invasion***	
		9-May-17						***nephrectomy + lymphadenectomy***	***histology of surgical sample***	***ccRCC + LN invasion***	
		22-Jun-17						***sunitinib (1st cycle)***			
			19-Jul-17	**1**	**nd**	**1**	***91***				
		28-Jul-17							***CT scan***		***PD***
		2-Aug-17						***sunitinib (2nd cycle)***			
		13-Sep-17						***sunitinib (3th cycle)***			
		6-Oct-17						***further sunitinib cycles***	***CT scan***	***reduction of lung lesions***	***PR***
			19-Oct-17	**2**	**2**	**nd**	***10***				
		1-Jan-18						***resection of brain metastasis***	***MRI***	***4.5 cm cerebral metastasis***	***PD***
		16-Jan-18	16-Jan-18	**0**	**0**	**nd**	***323***		***CT scan***	***increase of lung lesions***	***PD***
		29-Jan-18							***CT scan***	***lung carcinosis, visceral metastasis***	***PD***
			31-Jan-18	**1**	**0**	**nd**	***nd***				

nd = not done.

## Data Availability

The datasets used and/or analyzed during the current study are available from the corresponding author upon reasonable request.

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
