# Peer review of "Liquid Biopsy in Pediatric Renal Cancer: Stage I and Stage IV Cases Compared"

_diagnostics, 2020, doi:10.3390/diagnostics10100810_

Round 1

Reviewer 1 Report

The authors reported the change of CTCs and CECs in 2 pediatric renal cancer.

Because they reported only two cases, the scientific soundness might be low.

I know pediatric renal cancer is rare. However, the authors should show 5 cases at least to make their reports better.

Author Response

Comment 1

The authors reported the change of CTCs and CECs in 2 pediatric renal cancer.

Because they reported only two cases, the scientific soundness might be low.

I know pediatric renal cancer is rare. However, the authors should show 5 cases at least to make their reports better.

Replay 1:

We were aware that our study shows some limits, implicit of case reporting studies, namely few patients, and treatments, follow-up and timeline schedule of blood draws nonhomogeneous, because of different disease stage.

Despite these limits, we decided upon cases reporting, since our findings offer the rationale to further investigations of CTCs in larger pediatric cohorts. This rationale was lacking until now, because we are using here, for the first time, liquid biopsy in pediatric RCC.

Change in the text: The limits of our case report have been briefly commented in Discussion and Conclusion (lines 175-178).

Reviewer 2 Report

In this study Rossi et el. analysed the number of circulating tumor and endothelial cells in 2 patients with ccRCC, together with the expression of apoptosis marker M30 and tutor progression marker cMET.

It is an interesting study of relevance for future ability to stratify patients based on liquid biopsies methods; however the parameters are not measured at all time-point of follow-up or treatments and these analysis should be added, including levels at baseline (patient 1 ) to be able to draw a conclusion on the fluctuations of these circulating markers after treatment or to compare the changes with clinical outcomes.

Also it would be interesting to follow the fluctuations in CTC and CEC during the off-phase of treatment, as tumor evasion might occur in this time (as the authors suggested).

How do you explain that M30 levels are not decreased compared to baseline after treatment in patient 2? Please include the description and discussion of M30 changes in the text as well.

Would it be possible to add 2 patients at stage 2 and 3 to compare CTC, CEC, M20, CMET levels at baseline and after treatment with the rates in patients stage 1 and 4, and thus validate that these circulating marker reflect tumor growth and the gravity of the clinical situation?

Author Response

Comment_1

It is an interesting study of relevance for future ability to stratify patients based on liquid biopsies methods; however the parameters are not measured at all time-point of follow-up or treatments and these analysis should be added, including levels at baseline (patient 1 ) to be able to draw a conclusion on the fluctuations of these circulating markers after treatment or to compare the changes with clinical outcomes.

Reply 1: Because we were using here, for the first time, some liquid biopsy tools in renal pediatric cancer, to give approval, the Ethics Committee required to limit the schedule of blood draws to time-points of conceivable usefulness for the patients, in the view to minimize patients discomfort. In fact, we could not reduce the blood draw volume because of CTC rarity.

Therefore, in case-1, stage I, we planned to monitor the post-surgery period, in order to disclose early potential disease recurrence, as previously reported e.g. in breast cancer.

Conversely, in case-2, stage IV, the patients was evaluated at diagnosis of metastatic disease, because the scientific literature previously reported the prognostic value of CTCs, when determined at diagnosis of metastatic disease in several malignancies.

Change in the text: The limits of our case report have been briefly commented in Discussion and Conclusion (lines 175-178).

Comment 2

Also it would be interesting to follow the fluctuations in CTC and CEC during the off-phase of treatment, as tumor evasion might occur in this time (as the authors suggested).

Reply 2: We had already demonstrated that tumor evasion occurs during the off-phase of Sunitinib in adult ccRCC patients, by measuring CTC and CEC at the end of 4th week of treatment and then 2 weeks later, at the 1st week of the subsequent cycle; the results of the adult ccRCC study have been elsewhere published (Rossi E., et al BJC, 2012).

For the ethical reason exposed in response to comment 1, we could not employ this timeline schedule in pediatric ccRCC.

However, we previously reported in adult ccRCC that tumor-evasion mainly occurs in case of moderate and delayed response to anti- angiogenic drug. This unsatisfactory response to the treatment had been revealed by the slow increase of CEC number at the end of the on-phase of Sunitinib, between one cycle and the next one.

On these bases, when we had to reduce as low as possible the blood draw schedule for monitoring pediatric ccRCC, we chose to enumerate CTC and CEC at the end of the on-phase of the drug. In fact, the comparison of data obtained at this time point in subsequent cycles, offers equally evidence of the effect of the off-phase of the treatment.

As previously reported in adult ccRCC patients, in case 2, stage IV, the delayed increase of CEC was accompanied by the lack of decrease of the CTC count, since at the end of 4th cycle (October 2017) we found only 10 CECs/4 mL PB, and a positive CTC level (2 cells)…. Only at 7th cycles of Sunitinib we observed a strong response to treatment (323 CECs/4 mL PB), and CTC enumeration resulted negative; however, in the meantime, disease had progressed…

Changes in the text: The cycle number has been edit, and is now correctly indicated as 4th (line 163). Our previous demonstration of similar phenomenon in adult ccRCC has been cited (lines 167-168).

Comment 3

How do you explain that M30 levels are not decreased compared to baseline after treatment in patient 2? Please include the description and discussion of M30 changes in the text as well.

Reply 3: During the translation from Excel sheet to Word file in preparing the Table, the last line of the Excel sheet was lacking; the Editor can easily verify the mistyping of the Table during my first submission, looking at the original Word file that I uploaded at the journal submission platform. Clicking on the Table “open” and then “modify” can recover the original Excel sheet used for transferring in Word, and verifies that the last line of Excel sheet reports the result of 31 January test (1 CTC, M30-negative)

Actually, in the original main text, we had already commented (line 152) “the lack of apoptosis marker at the last CS test”…. A sentence that appeared inconsistent with the mistyped Table.

Changes in the text: The Table has been edited. We commented the M30 expression on CTCs and, to this purpose, calculated the ΔAUC parameter previously reported in adult ccRCC (lines 153-157).

Comment 4

Would it be possible to add 2 patients at stage 2 and 3 to compare CTC, CEC, M20, CMET levels at baseline and after treatment with the rates in patients stage 1 and 4, and thus validate that these circulating marker reflect tumor growth and the gravity of the clinical situation?

Reply 4:

In principle, it is not expected that CTC level reflect tumor growth, rather CTCs are an independent prognostic marker of disease condition, i.e. indolent vs. aggressive one, at least in metastatic breast cancer (Cristofanilli, Consensus paper, 2019). For the other markers we used here, there are even fewer data, about this.

Anyway, it needs a larger sample size to validate these markers ... In fact, CTC number < vs. > cut-off defines the probability of best vs. worst prognosis; to obtain this evidence, we need larger studies, ad hoc designed.

Hence, why we are cases reporting?

Despite the implicit limits of this kind of studies, we decided upon cases reporting, since our findings offer the rationale to further investigations of CTCs in larger pediatric cohorts. This rationale was lacking until now, because we are using here, for the first time, liquid biopsy in pediatric RCC.

Changes in the text: The limits of our case report have been briefly commented in Discussion and Conclusion (lines 175-178).

Round 2

Reviewer 1 Report

No comment

Reviewer 2 Report

All points arise by the reviewer have been addressed in the "author's reply" document. Although addition of original lacking data at certain time-points was not possible because of lack of primary material, the authors had made some changes in the text in response to the reviewer's questions.